# A Fluorescent Polyurethane with Covalently Cross-Linked Rhodamine Derivatives

**DOI:** 10.3390/polym12091989

**Published:** 2020-09-01

**Authors:** Saiqi Tian, Yinyan Chen, Yifan Zhu, Haojun Fan

**Affiliations:** 1College of Education, Wenzhou University, Wenzhou 325035, China; 18210411313@stu.wzu.edu.cn (Y.C.); 18210411148@stu.wzu.edu.cn (Y.Z.); 2National Engineering Laboratory for Clean Technology of Leather Manufacture, Sichuan University, Chengdu 610065, China; fanhaojun@scu.edu.cn; 3State Key Laboratory of Polymer Materials Engineering, Sichuan University, Chengdu 610065, China

**Keywords:** fluorescent, polyurethane, crosslink, rhodamine

## Abstract

Rhodamine derivatives (RDs) with three reactive hydrogens were synthesized and well characterized by Fourier transform infra-red spectroscopy (FTIR), ^1^H nuclear magnetic resonance (^1^H NMR) and electrospray ionization mass spectra (ESI mass). Then, the obtained RD was covalently cross-linked into polyurethane (PU) matrix through chemical linkages to fabricate a network structure, and the fluorescent properties, mechanical properties, thermal stability, and emulsion particle size were systematically investigated. Results demonstrate that PU-RD maintains initial fluorescent properties and emits desirable yellow fluorescence under ultraviolet irradiation. Moreover, compared with linear PU without fluorescers, PU-RD shows clearly improved mechanical properties and thermal stability, on account of the formed network structures.

## 1. Introduction

Recently, significant interest has been directed to the development of fluorescent polymers, because of their intrinsic advantages, including low toxicity, good biocompatibility, long-term stability and better processability [1,2,3]. In particular, fluorescent polyurethane (PU) is one of the most desirable polymers, owing to its outstanding combination of unusual features, such as excellent mechanical strength, good abrasion resistance, and high elasticity [4,5,6,7]. These facilities make the materials useful for technological applications in a very broad range of fields, such as coating materials, textiles, paper making, organic light emitting diodes (LEDs), and fluorescent probes. For example, Lian and coworkers [8] designed and prepared a fluorescent polyurethane nanocomposite via one-pot in situ synthesis methods. This the fluorescence of composites can be changed with various carbon precursors with different structures, showing wide potential application for flexible solid-state lighting and displays.

A fluorescent PU is mostly made of a mixture of fluorescers and polymer. However, physically doping fluorescers into PU matrices inevitably leads to some shortcomings, including poor mechanical properties and incompatibility with the matrix, attributed to weak interaction between fluorescers and PU [9]. Generally, PU structure is determined by raw materials, hard and soft segment, molecular weight, polydispersity, and cross-linking ability. It can be easily designed by changing the types and quantities of socyanate, polyol, surfactants, catalysts, fillers, and cross-linking agents during the manufacture process or via advanced characterization techniques, so as to meet the desired performance [10,11,12]. Thus, attention has been drawn to the chemical attachment of small molecular fluorescers into polyurethane backbones.

As commonly recommended coating materials, PU is supposed to exhibit good mechanical properties and ideal thermal stability [13,14,15]. As mentioned before, the final properties have the possibility to be tailored by changing the composition. In addition, PU properties highly depend on the chemical cross-linking of the polymer chains. It is regarded that moderate cross-linking can positively influence these performances [16,17]. Previous research has proved that the thermal stability of cross-linked polyurethanes is higher than that of the linear one [18,19]. Network structures also result in improvement of mechanical properties [20,21]. Some works have been devoted to the influence of crosslinker on properties of PU. For example, Arévalo-Alquichire et al. [22] synthesized a PU using pentaerythritol as a crosslinker. Results indicated that the largest concentration of cross-linkers produced more distributed segment segregation and higher thermal stability. Cross-linking process in PU could be carried out through various methods, wherein low-molecular-weight compounds with more than two reactive hydrogens were used as cross-linkers, such as trimethylolpropane (TMP) and triethanolamine (TEA) [23,24,25]. Moreover, rhodamine chromophores have advantages of long wavelengths of excitation (greater than 550 nm) and emission (590 nm), good photostability, and high fluorescent quantum yields. They are widely used fluorescent micromolecules [26]. However, the report of using rhodamine derivatives as polyurethane cross-linkers is still very scarce. Guided by these previous findings, we designed and synthesized rhodamine derivatives (RD) with three reactive hydrogens, which are ideal cross-linkers for PU. Subsequently, fluorescent polyurethanes (PU-RDs) were prepared by being covalently cross-linked by RD. The fluorescent properties, mechanical properties, thermal stability, and emulsion particle size of PU-RDs were fully investigated. Our results verify that PU-RDs exhibit very promising and unique properties. PU-RDs maintain appealing fluorescent properties after polymerization with RD, and emit yellow fluorescence under a ultra-violet (UV) lamp. As result of network structures, PU-RDs present higher mechanical properties and thermal stability than linear PU. The easy-to-obtain materials, facile preparation, good mechanical properties, and thermal stability make it suitable for potential practical applications.

## 2. Experiment

### 2.1. Materials

*1,4*-butanediol (BDO), *2,2*-bis (hydroxymethyl) propionic acid (DMPA), triethylamine (TEA) were purchased from Micxy Chemical Co., Ltd. (Chengdu, China). Isophorone diisocyanate (IPDI), rhodamine 6G, *2*-aminoethano, acetonitrile, and acetone were supplied by Aladdin Industrial Corporation (Shanghai, China). Poly(propylene glycol) with molecular weights of 2000 (PPG220) and polytetrahydrofuran glycol with molecular weights of 1000 (PTMG210) were sourced from Dow Chemical Company (Midland, MI, USA) and dried at 120 °C under high vacuum (0.5 mmHg) for 12 h before use. Bismuth neodecanoate was provided by Deyin Chemical Co. Ltd. (Shanghai, China).

### 2.2. Synthesis of Rhodamine *Derivative* (RD)

Rhodamine 6G (5 g, 10.4 mmol) was dissolved in 80 mL of acetonitrile. Thereafter, *2*-aminoethanol (1.85 mL, 31.3 mmol) was added to the solution. The reaction was stirred under reflux at 90 °C for 2 h (See Scheme 1). After that, the mixture was cooled to room temperature. The solid was filtered and washed with water three times, and dried under vacuum. Yield: 93%.

### 2.3. Synthesis of Fluorescent Waterborne Polyurethane (PU-RD)

IPDI, PPG220, PTMG210, and BDO (molar ratio IPDI: PPG220: PTMG210: BDO = 6:6:1:1.6) were poured into the four-necked separable reaction flask equipped with a mechanical stirrer, thermometer, nitrogen inlet, and condenser with a drying tube. The mixture was heated to 80 °C with a drop of bismuth neodecanoate (0.05 wt. %) as a catalyst, and the reaction was carried out for 2–3 h. Afterwards, the intermediate was chain extended by adding DMPA (molar ratio of IPDI: DMPA = 6:1.1) at that temperature for another 2 h to form a liner prepolymer. During the pre-polymerization, a suitable amount of acetone was added to dilute the viscosity of the reaction system. Successively, RD was added and the pre-polymerization was carried out at 80 °C under N_2_ atmosphere for a duration of 2 h. After that, the system was cooled to 50 °C. As a neutralization agent, TEA (molar ratio TEA: DMPA = 1.1:1) was then added to react with the carboxylic group. Ultimately, the resultant mixture was dispersed by appropriate deionized water with vigorous stirring conditions for 1.5 h. The final PU-RD was obtained, and the solid content of the emulsion was approximately 30%. Samples here are abbreviated as PU-RD-x, and the corresponding RD weight concentrations of PU-RD-1, PU-RD-2, PU-RD-3, and PU-RD-4 are 0.5%, 1.0%, 1.5%, and 2.0%, respectively.

As a control, linear polyurethane (PU) was synthesized using the same procedure without the presence of RD.

### 2.4. Measurements

Fourier transform infra-red (FTIR) spectra were obtained with a Nicolet IS10 FTIR spectrometer (ThermoFisher Scientific, Waltham, MA, USA) in the range from 400 to 4000 cm^−1^ with 32 scans at 2 cm^−1^ resolution. ^1^H nuclear magnetic resonance (^1^H NMR) spectra were acquired on a AV400 NMR (400 MHz; Bruker, Ettlingen, Germany) spectrometer operating in the Fourier transform mode at 30 °C with deuterated dimethyl sulfoxide (DMSO-d_6_) as the solvent and tetramethylsilane as the internal reference. Electrospray ionization (ESI) mass spectra were obtained on a Finnigan LCQ Advantage ion trap mass spectrometer (ThermoFisher Corporation). The fluorescent emission properties were measured at room temperature using a spectrophotometer (Fsp920, Edinburgh Instruments, Livingston, UK) and the slit was 1 nm in width. The DMA measurements were conducted in the three-point mode with DMA Q800. Tensile tests (tensile strength and elongation at break) were carried out on the samples with a universal material testing machine (model tensiTECH) supplied by Tech-Pro Inc. (Woodstock, NY, USA) at room temperature. The samples were cut into dumb-bell shapes with a thickness of around 1.0 mm and the crosshead speed was set at 10 mm min^−1^. For each data point, five samples were tested to obtain an average value. Thermogravimetric analysis (TGA) was performed with a TG-209 F1 thermal analyzer (Netzsch, Germany) at a scanning rate of 10 °C min^−1^ under a nitrogen flow rate of 60 mL min^−1^. Particle sizes were obtained by laser particle size analyzer (Mastersizer 3000E).

## 3. Results and Discussion

### 3.1. Structure Characterization

The chemical structure of RD was analyzed by FTIR spectroscopy, ^1^H NMR, and ESI mass. Figure 1 presents FTIR spectra of rhodamine 6G and RD. After modified by *2*-aminoethanol, the FTIR peaks at 2363 cm^−1^ and 2342 cm^−1^, which are assigned to NH^+^ in rhodamine 6G, disappear. Moreover, the peak of –C=O distinctly blue-shifts from 1716 cm^−1^ to 1682 cm^−1^, on account of the transformation from ester linkage (–COOC–) [27] to amido linkage (–CO–N–) [28]. Figure 2 displays digital images of rhodamine 6G and RD in daylight. The color of powders is obviously changed after reaction. The ^1^H NMR spectrum of RD is depicted in Figure 3, and we can find the corresponding peaks: δ 7.94 (m, 1H), 7.48 (m, 2H), 6.97 (m, 1H), 6.22 (s, 2H), 6.16 (s, 2H), 3.32 (s, 2H), 3.14 (m, 2H), 2.50 (s, 4H), 1.23 (m, 6H). Furthermore, the mass of RD (C_28_H_32_O_3_N_3)_ is calculated to be 458.58. As presented in Figure 4, the ESI mass measured mass is 458.24. All these results indicate the successful synthesis of RD.

PU-RD-3 was chosen as a representative example to confirm the structure of PU-RD by FT-IR spectrometry, as shown in Figure 5. We can find the typical peaks of PU, including those at 3334 cm^−1^ (N–H stretching), 2970 and 2931 cm^−1^ (–CH_3_ and –CH_2_– stretching), 1705 cm^−1^ (C=O group in urethane) and 1108 cm^−1^ (C–O–C stretching) [29]. The characteristic absorption peak of semicarbazide formed from the reaction of –NH– in RD and –NCO in IPDI is observed at 1673 cm^−1^ in the spectrum of PU-RD, indicating that RD has been successfully implanted into the polyurethane main chains [30].

### 3.2. Fluorescent Properties

To evaluate the fluorescent properties of PU-RDs, their fluorescence spectra were acquired. The fluorescence spectra of RD, PU, and PU-RDs, assessed by the wavelengths 365 nm at 25 °C, are displayed in Figure 6. It can be seen that PU without RD unit shows no emission peaks, while RD and PU-RDs show similar emission shape and the maximum emission wavelength. This demonstrates that the PU-RDs still maintain appealing fluorescence after polymerization. After being attached into PU networks, the local environment around the RD fluorophores is changed [31]. Consequently, compared to the spectrum of RD, the maximum peaks of PU-RDs shift from 560 nm to 556 nm and show a hypochromatic shift. In addition, the maximum intensity of PU-RDs gradually increases as the content of RD increases.

Figure 7 depicts images of PU and PU-RDs films taken both in daylight and in darkness under 365nm ultraviolet irradiation. We can see that PU-RDs present red in daylight. In darkness, PU without RD shows no light emission. It is quite clear that PU-RDs emit yellow fluorescence under a 365 UV lamp.

### 3.3. Mechanical Properties

Tensile tests and dynamic thermomechanical analysis (DMA) were performed to measure the mechanical properties of PU and PU-RDs, and the results are summarized in Table 1. Figure 8 displays stress–strain curves of PU and PU-RDs. As the content of RD increases, the tensile strength enhances and the elongation at break weakens, owing to the cross-linking of polyurethane by –OH and two –NH– of RD. The dissipation factor (tan δ) curves as a function of temperature for the PU and PU-RDs are given in Figure 9, where the peak of tan δ is defined as the glass-transition temperature (T_g_) [32,33]. The higher cross-linking density also restricts polymer-chain mobility, leading to the increase of glass-transition temperature [34,35].

### 3.4. Thermal Stability

The thermal stability of polyurethanes is a significant parameter for their application as a coating material. PU-RD-3 was chosen as a representative example to study the thermal stability. TGA and differential thermogravimetric (DTG) curves of PU and PU-RD-3 are given in Figure 10a. The corresponding data are briefly summarized in Table 2. In general, the TGA curves for the two samples are similar. PU-RD-3 displays slightly higher thermal stability. There is a typical three-step decomposition process. The first step under 300 °C is ascribed to the evaporation of acetone and water [36]. The initial weight loss (temperature taken at the 5 wt. % of weight loss, T_5%_) for PU and PU-RD-3 appears at 279.2 and 281.9 °C, respectively. The second step corresponds to the decomposition of hard segments i.e., urethane groups [37]. T_max1_ of PU and PU-RD-3 is at 330.4 and 346.6 °C, respectively. PU-RD-3 is cross-linked by RD, and the network structures present higher thermotolerance [35]. Moreover, the RD, composed of three benzene rings, is supposed to exhibit good thermal stability and is difficult to decompose with increasing temperature [38,39]. Both the network structures and the attachment of benzene units as hard segments result in a higher T_max1_. The third step is related to soft segments (ether or ester bond) [40]. The weight-loss curves are similar during this step. Therefore, PU-RD has good stability in the storage and application.

The fluorescent properties are also expected to be stable after thermal exposure. PU-RD-3 was kept at 100 °C for 24 h in an oven. Then, the fluorescent emission spectrum was obtained. From the fluorescent emission spectra in Figure 10b, we can see that the maximum peak and shape of PU-RD are almost the same before and after thermal exposure, demonstrating desirable stability.

### 3.5. Emulsion Particle Size Analysis

Figure 11 depicts the particle size distribution of PU and PU-RDs emulsions, respectively. The detailed data for average size and particle dispersion index are summarized in Table 3. The average size of PU is 48.20 nm. Regarding PU-RD-4, it is measured to be 78.82 nm. The cross-linkers are copolymerized with PU chains, forming network structures. Therefore, the molecular chains get stiffer, leading to poorer hydrophilicity. This is unfavorable for emulsification and dispersion [41]. Consequently, the average sizes gradually decrease as the content of RD increases. Furthermore, with the incorporation of RD, the particle dispersion index gets narrower. With more network structure formed during the polymerization, more molecules are intertwined around major structures, as described in Figure 12.

## 4. Conclusions

In summary, we designed and synthesized fluorescent polyurethanes (PU-RDs) that were covalently cross-linked by rhodamine derivatives (RD). First, RD with three reactive hydrogen was successfully achieved. Afterwards, it was attached into polyurethane chains as cross-linkers. RD and PU-RDs show similar emission shape and the maximum emission wavelength, indicating the fluorescence is still maintained after polymerization. PU-RDs present red in daylight and emit yellow fluorescence under a 365 UV lamp. Due to the cross-linking of polyurethane by –OH and two –NH– of RD, PU-RDs form network structures. The glass-transition temperature, tensile strength, thermal stability, and average size of emulsions increase as the content of RD increase. Because of these unique properties, as well as the facile and convenient preparation method, this fluorescent polyurethane shows a wide application prospect in coating materials, textiles, anti-fake labels, organic LEDs, and fluorescent probes.

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
