# Peer review of "A Fluorescent Polyurethane with Covalently Cross-Linked Rhodamine Derivatives"

_polymers, 2020, doi:10.3390/polym12091989_

Round 1

Reviewer 1 Report

The authors have synthesized Rhodamine derivatives and examined them using NMR and FTIR spectroscopic techniques, among others. They found that the network structures of PU-RD exhibit improved mechanical properties and thermal stability. The study is devotedly conducted, but the standard of English is no good. There are several grammatical mistakes scattered all around the ms.

Some major concerns are given below:

  1. The fluorescent properties, mechanical properties, thermal stability and emulsion particle size of PU-RDs was detailly

Concern: Was should be were? I don’t know of the word “detailly”. It is a word in the English dictionary? Did they mean “…fully investigated”?

  1. Guided by these previous findings, we designed and synthesized rhodamine derivatives (RD) with three reactive hydrogen, which is ideal to be crosslinkers for PU

Concern: which IS/ARE ideal to be crosslinkers for PU? Is it three reactive hydrogen (or hydrogens)?

  1. While assigning the FTIR-based vibrational modes, it is recommended to compare them with literature. I cannot see it in the current version of the ms.
  2. 1.Structure characterization

Concern: Is it a structure or vibrational characterization?

  1. Significantly improved discussion is necessary in the results and discussion section, which is lacking in the current version – providing pictures and assigning some vibrational modes and others are not really science.
  2. References are not up to date.

In conclusion, I am aiming to see a significantly improved ms for further review

Author Response

Response to Reviewer 1 Comments

Dear reviewer,

Thanks for your comments concerning our manuscript entitled A fluorescent polyurethane with covalently crosslinked rhodamine derivatives”. Those comments are all valuable and helpful for improving our paper. Revisions are highlighted using the "Track Changes" function in Microsoft Word. The main corrections in the paper and the responds to the your comments are as following:

Point 1:

The fluorescent properties, mechanical properties, thermal stability and emulsion particle size of PU-RDs was detailly

Concern: Was should be were? I don’t know of the word “detailly”. It is a word in the English dictionary? Did they mean “…fully investigated”?

Response: We are sorry for bothering you with this kind of mistakes that we should have avoided. We have reviewed and re-checked throughout the whole manuscript. Spelling mistakes, syntactic, as well as grammatical errors have been eliminated wherever possible in our revised version to address this issue. Thanks.

Point 2:

Guided by these previous findings, we designed and synthesized rhodamine derivatives (RD) with three reactive hydrogen, which is ideal to be crosslinkers for PU

Concern: which IS/ARE ideal to be crosslinkers for PU? Is it three reactive hydrogen (or hydrogens)?

Response: Thanks for your kind question. “is” is corrected as “are”. “hydrogen” is corrected as “hydrogens”.

Point 3:

While assigning the FTIR-based vibrational modes, it is recommended to compare them with literature. I cannot see it in the current version of the ms.

Response: We are so grateful for this kind suggestions. New references were added to approve FTIR-based vibrational modes. (R. 27 and R. 28)

Point 4:

1.Structure characterization

Concern: Is it a structure or vibrational characterization?

Response: Thank you for your question. It is a structure characterization.

Point 5:

Significantly improved discussion is necessary in the results and discussion section, which is lacking in the current version – providing pictures and assigning some vibrational modes and others are not really science.

Response: Thanks for pointing out the insufficiency of the manuscript. Fluorescent PU has been widely used as coating materials. Physically doping fluorescers into PU matrices inevitably leads to some shortcomings, including poor mechanical properties and incompatibility with the matrix. So we try to achieve the chemical attachment of small molecular fluorescers (RD) into polyurethane backbones. FTIR spectra and all the properties influenced by network structures can demonstrate the chemical combination of RD with PU. Particularly, both mechanical property and thermal stability are of great importance to coatings materials. The structure of fluorescent polyurethane are normally linear. It has been observed that linear PUs exhibited inappropriate mechanical properties such as dumping, low elongation, etc. Network structures gives the result in improvement of mechanical properties. Previous research has also proved that the thermal stability of crosslinked polyurethanes is higher than that of the linear one. For this purpose, we designed and synthesized crosslinkers based on rhodamine derivatives. The mechanical properties and thermal stability were investigated. The crossliners are chemically incorporated in to PU chains and offer PU fluorescent properties. Moreover, because of the network structures formed in PU-RD, the mechanical properties and thermal stability is also enhanced, compared with the linear one. To further develop our article, we have added more discussions in the results and discussion section. Thanks again for your kind suggestion.

Point 6:

References are not up to date.

Response: Thanks for your question. Some references are updated.

We tried our best to improve the manuscript and made some changes in the manuscript.  These changes will not influence the content and framework of the paper. We appreciate for reviewers’ warm work earnestly, and hope that the correction will meet with approval.

Once again, thank you very much for your comments and suggestions.

Reviewer 2 Report

  1. In introduction part, the literature reviews related with the cross-linking agents for fluorescent PU (e.g., kinds and cross-linking properties on fluorescent PU material properties) should be reinforced to improve clearity of this article. In addition, in same manner, the objective of this article should be described more clearly in this part. 
  2. The applications of fluorescent polymer need to be described. 
  3. The information about rhodamine derivatives should be described in the introduction part. 
  4. The FT-IR properties of base material, PU, PU-rodamine 6G, and PU-RD should be additionally determined and compared. Also, NMR and ESI results of RD need to be compared with the original material (rodamine 6g). 
  5. How different fluorescent properties between PU-RD and PU-rodamine 6G? 
  6. Are the fluorescent properties of PU-RD maintained after thermal exporsure? 
  7. What is advantage of developed material compared with other similar fluorescent materials?
  8. Authors can suggest the application areas of developed materials more detail. 

Author Response

Response to Reviewer 2 Comments

Dear reviewer,

Thanks for your comments concerning our manuscript entitled A fluorescent polyurethane with covalently crosslinked rhodamine derivatives”. Those comments are all valuable and helpful for improving our paper. Revisions are highlighted using the "Track Changes" function in Microsoft Word. The main corrections in the paper and the responds to the your comments are as following:

Point 1:

In introduction part, the literature reviews related with the cross-linking agents for fluorescent PU (e.g., kinds and cross-linking properties on fluorescent PU material properties) should be reinforced to improve clearity of this article. In addition, in same manner, the objective of this article should be described more clearly in this part.

Responds: We appreciate the reviewer for this kind recommendation. We added more references related to cross-linking agents for fluorescent PU and re-wrote introduction part to describe it more clearly. I am sorry for not being able to highlight the objective of this manuscript in the early version. In the revised version, I have added more descriptions to objective of this work. Fluorescent PU has been widely used as coating materials. Physically doping fluorescers into PU matrices inevitably leads to some shortcomings, including poor mechanical properties and incompatibility with the matrix. So we try to achieve the chemical attachment of small molecular fluorescers (RD) into polyurethane backbones. FTIR spectra and all the properties influenced by network structures can demonstrate the chemical combination of RD with PU. Particularly, both mechanical property and thermal stability are of great importance to coatings materials. The structure of fluorescent polyurethane are normally linear. It has been observed that linear PUs exhibited inappropriate mechanical properties such as dumping, low elongation, etc. Network structures gives the result in improvement of mechanical properties. Previous research has also proved that the thermal stability of crosslinked polyurethanes is higher than that of the linear one. For this purpose, we designed and synthesized crosslinkers based on rhodamine derivatives. The mechanical properties and thermal stability were investigated. The crossliners are chemically incorporated in to PU chains and offer PU fluorescent properties. Moreover, because of the network structures formed in PU-RD, the mechanical properties and thermal stability is also enhanced, compared with the linear one. Thanks again for your kind suggestion.

Comment 2:

The applications of fluorescent polymer need to be described.

Responds: Thank you for your suggestions. We have add more description of applications in introduction part.

Comment 3:

The information about rhodamine derivatives should be described in the introduction part.

Responds: We appreciate this helpful suggestion. Information about rhodamine derivatives has been added in the introduction part.

Comment 4:

The FT-IR properties of base material, PU, PU-rodamine 6G, and PU-RD should be additionally determined and compared. Also, NMR and ESI results of RD need to be compared with the original material (rodamine 6g).

Responds: We thank the reviewer for this helpful recommendation. FRIT spectra of PU, PU-RD are added in manuscript (see Figure 2). After incorporated of RD into PU, a new peak at 1673 cm-1 appears, which attributes to semicarbazide formed from the reaction of –NH- of RD and -NCO of IPDI. This could give strong evidence that RD has been successfully implanted into the polyurethane main chains. Rhodamine 6G is a common fluorescent materials. We can easily find its NMR and ESI results in references. Through H1 NMR spectrum of RD, we can find all the corresponding peaks. ESI spectrum of RD presents the mass of measured compound, and the result exactly ascribes to the calculated mass of RD. Therefore, combined with the FTIR spectra and color change of powders, we think the H1 NMR and ESI results of RD can confirm its structure.

Comment 5:

How different fluorescent properties between PU-RD and PU-rodamine 6G?

Responds: We appreciate your question. PU-RD presents the same maximum emission peak and shape as RD. PU-rhodamine 6G presents the same maximum emission peak and shape as rhodamine 6G. The main difference between PU-RD and PU-rhodamine 6G is structure. PU-RD is network, while PU-rhodamine 6G is linear. The difference in structure results in diverse mechanical properties and thermal stability.

Comment 6:

Are the fluorescent properties of PU-RD maintained after thermal exporsure?

Responds: We thank the reviewer for this kind question, and this really helps us to modify our manuscript. The fluorescent properties of PU-RD are still maintained after thermal exposure. Moreover, we have added fluorescent emission spectra of PU-RD before and after thermal exposure in the manuscript (see Figure 7(b)).

Comment 7:

What is advantage of developed material compared with other similar fluorescent materials?

Responds: We thank the reviewer for raising this question. Normally, the structure of fluorescent polyurethane are linear. It has been observed that linear PUs exhibited inappropriate mechanical properties such as dumping, low elongation, etc. (Yarmohammadi, Propellants Explos. Pyrotech. 2018, 43, 1–7) PU-RD with network structures possesses developed mechanical properties, compared with the linear one. Besides, thermal stability of PU-RD is also enhanced. PU is widely used as coating materials. Both mechanical property and thermal stability are important to coatings materials.

Comment 8:

Authors can suggest the application areas of developed materials more detail.

Responds: We appreciate this helpful suggestion. We have added promising application areas of this developed materials in this manuscript.

We tried our best to improve the manuscript and made some changes in the manuscript.  These changes will not influence the content and framework of the paper. We appreciate for reviewers’ warm work earnestly, and hope that the correction will meet with approval.

Once again, thank you very much for your comments and suggestions.

Round 2

Reviewer 1 Report

The authors have considered all of my comments. They have revised their paper. It advanced with the discussions made, the picture qualities, and references. I think the paper is almost complete for possible publication in the journal.

Author Response

Thank you for your encouragement to our work. We would like to express our great appreciation to your warm work earnestly.

Reviewer 2 Report

The authors responded enoughly to my questions. This revised manuscript could be published in this journal after final check minor errors. 

Author Response

(The authors gave the same response as above.)
